

# MealMaster
## Aplikacja mobilna wspierająca zbilansowaną dietę oraz walkę z marnowaniem żywności

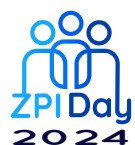

**Autorzy**: Julia Knura⊙ · Sebastian Kulessa⊙ · Jakub Nnoli⊙ · Justyna Stochniel⊙

**Opiekun:** Bogumiła Hnatkowska

### Streszczenie

W 2022 roku zmarnowano globalnie ponad miliard ton żywności, z czego 60% zostało wyrzucone w gospodarstwach domowych [14]. MealMaster może zmniejszyć tę liczbę poprzez ułatwienie zarządzania posiadaną żywnością i promowanie świadomych zakupów. Opracowana przez nasz zespół aplikacja mobilna dodatkowo ułatwia dobór odpowiednich dań dzięki możliwości przeglądania przepisów z uwzględnieniem danych o ich składnikach oraz spersonalizowanym propozycjom opartym o posiadane przez użytkownika produkty spożywcze. Jednocześnie dostarcza ona bezpośrednich korzyści dla użytkownika poprzez zmniejszenie kosztów zakupów spożywczych, skrócenie czasu potrzebnego na wybór posiłku oraz zniwelowanie uczucia zmęczenia częstymi wyborami (ang. *decision fatigue*). Aby dodatkowo zachęcić do korzystania z tego produktu, aplikacja pomaga w podejmowaniu świadomych decyzji dotyczących wartości odżywczych prowadzonej przez użytkownika diety oraz zróżnicowanie jej dzięki możliwości wypróbowania nowych dań.

## 1 WSTĘP

### 1.1 Problem marnowania żywności w gospodarstwach domowych

Marnowanie żywności jest problemem o dużym znaczeniu i skali. W 2022 roku, tylko na poziomie sprzedaży, branży gastronomicznej i gospodarstw domowych, zmarnowano ponad miliard ton żywności, czyli 19% dostępnej dla konsumentów żywności w skali globalnej [14]. Mimo że jedna trzecia ludzkości nie ma stabilnego dostępu do żywności, każdy człowiek rocznie wyrzuca średnio 79 kg jedzenia. Dodatkowo, straty w żywności w trakcie całego procesu produkcyjnego - od zbiorów do zużycia przez końcowego konsumenta - generują do 10% globalnej emisji gazów cieplarnianych. To prawie 5 razy więcej niż sektor lotniczy.

Aby zbudować rozwiązanie, mogące pomóc zmniejszyć ten problem, należy zrozumieć z jakich powodów jest wyrzucane jedzenie. Nasz zespół skupił się na gospodarstwach domowych - przyczyniają się one do 60% marnowanej żywności [14]. Według raportu przygotowanego w 2007 dla Waste and Resources Action Programme (w skrócie WRAP), można wyróżnić 3 grupy osób marnujących najwięcej jedzenia [13]:

- młode osoby (16-34 lat) pracujące na pełen etat,

- młode rodziny - rodzice w wieku 25-44 lat, pracujący lub pozostający w domu, z dziećmi poniżej 16 roku życia,

- ludzie mieszkający w mieszkaniach socjalnych, głównie pracujący fizycznie lub bezrobotni.

Na podstawie ankiety przeprowadzonej w ramach raportu, wyróżniono 7 kluczowych powodów marnowania żywności:

- kupowanie zbyt dużej ilości żywności,

- kupowanie żywności o krótkim okresie zdatności do spożycia, takich jak owoce i warzywa, w ramach podejmowania prób zdrowszego odżywiania i eksperymentowania z jedzeniem,

- zużywanie żywności w nieodpowiedniej kolejności (bez uwzględnienia ich okresu zdatności) i impulsywne wybieranie produktów do zjedzenia w pierwszej kolejności,

- porywcze sezonowe sprzątanie szafek, lodówek i zamrażarek ze starej, zapomnianej lub niechcianej żywności,

- wysoce restrykcyjne podejście do higieny żywieniowej i sugerowanych dat zdatności produktu do spożycia,

- sporządzanie zbyt dużej ilości jedzenia,

- niezadowolenie ze smaku jedzenia - głównie przez dzieci.

Warto przy tym odnotować, że młode rodziny głównie wyrzucają przygotowane przez siebie dania z powodu ugotowania zbyt dużej ilości lub niezadowolenia dzieci. Częściej też silnie trzymają się sugerowanych dat na etykietach żywności. Natomiast młode osoby pracujące głównie marnują zakupione produkty (zarówno otwarte, zużyte częściowo, jak i zamknięte). Często kupują za dużo lub impulsywnie, a także deklarują przygotowywanie zbyt dużej ilości jedzenia. W przeciwieństwie do młodych rodzin, mają mniej restrykcyjne podejście do spożywania żywności po upływie sugerowanego na etykiecie okresu zdatności.

## 1.2 Aplikacja jako rozwiązanie problemu

Sztokholmski Instytut Środowiska wskazuje, że rozwiązania technologiczne mogą odgrywać kluczową rolę w rozwiązaniu problemu marnowania żywności, jednak duży wpływ na ich sukces ma kwestia, czy ludzie będą chcieli zmienić sposób, w jaki żyją [12]. Według raportu WRAP można podzielić ludzi na 4 grupy w zależności od tego, jak bardzo przejmuje ich przedstawiony wyżej problem [13]:

- 13% ankietowanych przejmuje się problemem w wysokim stopniu i podejmuje znaczące wysiłki aby zredukować ilość marnowanej żywności,

- 28% ankietowanych przejmuje się problemem, ale rzadziej podejmuje działania w celu zredukowania ilości marnowanej żywności,

- 26% ankietowanych przejmuje się problemem w znikomym stopniu i nie podejmuje działań w celu zredukowania ilości marnowanej żywności, ale nie jest przeciwna takim działaniom,

- 33% ankietowanych nie przejmuje się problemem, a niektórzy z nich są przeciwni podejmowaniu działań mających na celu zredukowanie ilości marnowanej żywności.

Stworzenie przyjemnego doświadczenia dla użytkownika i wniesienie bezpośrednich korzyści innych niż zmniejszenie ilości wyrzucanego jedzenia wydaje się więc kluczowe, aby zaangażować część społeczeństwa niezainteresowaną problemem. Aplikacja adresująca problemy powodujące nie tylko marnowanie żywności, ale także inne, odczuwalne przez użytkownika konsekwencje, mogłaby nawet zaktywizować ostatnią z wymienionych grup ankietowanych. Okazuje się, że te same działania, które minimalizują ilość wyrzucanej żywności, prowadzą także do pozytywnie odbieranych przez użytkowników skutków. Wybrani przez Sztokholmski Instytut Środowiska przedstawiciele grup: młodych rodzin i młodych osób pracujących wskazują oszczędność pieniędzy i prowadzenie zbilansowanej diety jako jedne z najważniejszych czynników motywujących do oszczędzania żywności [12].

## 1.3 Zaproponowane rozwiązanie

Biorąc pod uwagę wyżej przedstawione dane, nasz zespół postanowił stworzyć aplikację mobilną skierowaną głównie do grupy młodych osób pracujących, ze względu na ich duży udział w marnowaniu żywności. Te osoby odczuwają trudności w kilku kluczowych obszarach powodujących marnowanie żywności i wskazują swój styl życia jako barierę uniemożliwiającą efektywne planowanie jadłospisu [13]. Jednocześnie, ze względu na swój wiek, są bardziej skłonne do sięgania po technologie w celu rozwiązania swoich problemów. Aby zachęcić do zmniejszenia ilości wyrzucanej żywności osoby niezainteresowane bezpośrednio tym problemem, aplikacja powinna odpowiadać na ich potrzeby i być łatwa w obsłudze.

Główne cele aplikacji to:

- ułatwienie zarządzania posiadanymi produktami spożywczymi,

- wspomaganie rozsądnych zakupów,

- ułatwienie planowania jadłospisu,

- ułatwienie prowadzenia zróżnicowanej diety.

Cele te odpowiadają na wybrane kluczowe problemy powodujące marnowanie żywności:

- kupowanie zbyt dużej ilości żywności,

- kupowanie żywności o krótkim okresie zdatności do spożycia w ramach podejmowania prób zdrowszego odżywiania i eksperymentowania z jedzeniem,

· zużywanie żywności w nieodpowiedniej kolejności,

· sezonowe sprzątanie szafek, lodówek i zamrażarek ze starej, zapomnianej lub niechcianej żywności,

· sporządzanie zbyt dużej ilości jedzenia.

Jednocześnie, obrane cele przynoszą korzyści, które ankietowani wskazali jako motywujące do minimalizowania ilości wyrzucanej żywności - oszczędzanie pieniędzy i zróżnicowanie diety. Osiągnięcie tych celów uprości także codzienne obowiązki związane z żywnością, takie jak planowanie zakupów i obiadów, co pozwoli użytkownikowi oszczędzić czas i zmniejszyć liczbę drobnych decyzji podejmowanych w trakcie dnia.

## 2 PRACE POWIĄZANE

### 2.1 Krótki przegląd istniejących rozwiązań. Analiza produktów konkurencyjnych

Zarówno na rynku polskim, jak i zagranicznym, istnieją już aplikacje mobilne, które rozwiązują niektóre aspekty problemu marnowania żywności.

Pierwszą kategorię takich produktów stanowią aplikacje skupiające się na redukowaniu marnowania żywności przez branżę gastronomiczną i sklepy spożywcze, m.in. Too Good To Go [11] i Foodsi [4]. Dzięki podziałowi użytkowników na dwie grupy - na przedsiębiorstwa oraz osoby prywatne - zapewniają one platformę, na której firmy mogą wystawiać oferty sprzedaży żywności zagrożonej zepsuciem po obniżonej cenie. Zachęcenie w ten sposób użytkowników indywidualnych zwiększa szansę na sprzedaż tych produktów przed upływem daty ważności. Podejście to jednak nie eliminuje całkowicie problemu marnowania żywności — rozwiązuje tylko jeden z jego etapów. Nie ma gwarancji, że użytkownicy nie kupią więcej, niż są w stanie spożyć. Ponadto, aplikacje tego typu zapewniają jedynie ograniczone możliwości śledzenia alergenów i przestrzegania diet.

Drugą kategorię stanowią aplikacje umożliwiające dzielenie się nadmiarem jedzenia między osobami mieszkającymi w bliskim sąsiedztwie, takie jak Olio [7] czy Taste and Share [10]. Użytkownicy mogą zamieszczać oferty żywności, którą chcą oddać, albo przeglądać dostępne w aplikacji darmowe produkty. Jednak poprawne działanie takich aplikacji wymaga aktywnego zaangażowania znacznej liczby osób zlokalizowanych w pobliżu. Prawdopodobnie nie zachęcą użytkowników, którzy nie zwracają uwagi na problem marnowania żywności — łatwiej jest im wyrzucić nadmiarowe produkty po ich zepsuciu niż monitorować ich stan i zamieszczać oferty w aplikacji.

Trzecią kategorię stanowią aplikacje wspierające użytkowników w zarządzaniu żywnością, którą już posiadają w swoich domach, np. NoWaste [6] czy Kitche [5]. Umożliwiają one tworzenie list zawartości lodówki, zamrażarki i spiżarni, przypisując do produktów ilość oraz termin zdatności do spożycia. Ułatwiają monitorowanie zapasów, wskazują produkty wymagające spożycia w pierwszej kolejności, tworzą listy zakupów, a czasem oferują przepisy, które można przyrządzić korzystając z posiadanych składników. Aplikacje należące do tej kategorii adresują problem marnowania żywności tam, gdzie jest on największy - w gospodarstwach domowych.

Produkt naszego zespołu należy do trzeciej kategorii. Skupia się na wspieraniu użytkowników w codziennych wyzwaniach, takich jak zarządzanie posiadaną żywnością, planowanie zakupów, oszczędzanie oraz podejmowanie decyzji dotyczących przygotowywanych posiłków. Ponadto dostarcza informacji na temat składników odżywczych oraz kaloryczności przepisów, co pozwala na lepsze monitorowanie nawyków żywieniowych pod kątem zdrowego stylu życia. Dzięki temu użytkownicy, nawet jeśli nie mają na celu ograniczania marnowania żywności, przyczyniają się do tego pośrednio. Dodatkowo, aplikacje należące do tej kategorii są stosunkowo mało popularne na polskim rynku, a czasami nawet niedostępne. Nasz produkt wypełnia tę lukę.

### 2.2 Główne założenia projektowe

Zdecydowaliśmy się na realizację projektu w formie aplikacji mobilnej połączonej z centralnym serwerem. Taka architektura pozwala użytkownikowi końcowemu korzystać z dowolnego urządzenia mobilnego, zapewniając szybki, łatwy i wygodny dostęp do funkcjonalności. Przechowywanie większości danych oraz wykonywanie kosztownych obliczeń odbywa się po stronie serwera, co odciąża aplikację kliencką i umożliwia łatwą synchronizację danych po aktualizacjach bazy. Aplikacja działa również w trybie offline, co pozwala na korzystanie z niej w sytuacjach, gdy użytkownik nie ma dostępu do internetu, np. podczas zakupów. Wybrane dane, takie jak lista posiadanych produktów czy ulubione przepisy, są przechowywane lokalnie, co zapewnia szybki dostęp bez potrzeby łączenia się z serwerem.

Wybraliśmy formę natywnej aplikacji Android, którą zrealizowaliśmy w języku Kotlin [1]. Na potrzeby serwera zastosowaliśmy framework Java Spring Boot [9], a do stworzenia aplikacji webowej dla administratora danych — framework React [8]. Technologie te zostały wybrane, ponieważ członkowie zespołu mieli już doświadczenie w ich użyciu, co pozwoliło na efektywne wykorzystanie ograniczonego czasu przeznaczonego na realizację projektu (10 tygodni). Hosting serwera został zrealizowany w ramach usługi AWS, którego koszty pokryły środki dostępne do dyspozycji w ramach konta studenckiego AWS Learners Lab.

Aplikacja wymagała bazy przepisów kulinarnych. Ponieważ nie znaleźliśmy odpowiedniej bazy zawierającej wszystkie potrzebne informacje, zdecydowaliśmy się na stworzenie własnej przez zastosowanie automatycznego pozyskiwania danych ze stron internetowych (ang. *web scraping*). Służące do tego skrypty zostały napisane w Pythonie z użyciem biblioteki Beautiful Soup [3]. Wybrane źródło - blog kulinarny BBC Good Food [2] - oferowało przepisy bogate w potrzebne nam informacje o składnikach odżywczych. Ze względu na ograniczenia czasowe zdecydowaliśmy się na gromadzenie przepisów w języku angielskim, unikając problemów z odmianą gramatyczną w języku polskim, np. w przypadku skalowania liczby porcji. W konsekwencji, by zachować spójność, język angielski stał się również językiem interfejsu użytkownika aplikacji mobilnej. Wersja demonstracyjna aplikacji zawiera ponad 300 przepisów, co stanowi kompromis między potrzebą obszernej bazy a koniecznością ręcznej weryfikacji danych.

Zakres produktów, które użytkownik może wprowadzać do aplikacji, został ograniczony do składników występujących w bazie przepisów. Zamknięta lista umożliwia unikanie błędów, takich jak literówki czy niewłaściwe odmiany, co uniemożliwiłoby identyfikację produktu.

## 3 REZULTATY

W wyniku tej pracy została stworzona aplikacja mobilna o nazwie MealMaster. Rysunki od 1 do 3 przedstawiają zrzuty ekranu z wybranymi widokami. Mając na uwadze postawione przed naszym zespołem cele, jak i charakterystykę grupy docelowej aplikacji, zostały zaimplementowane następujące funkcjonalności.

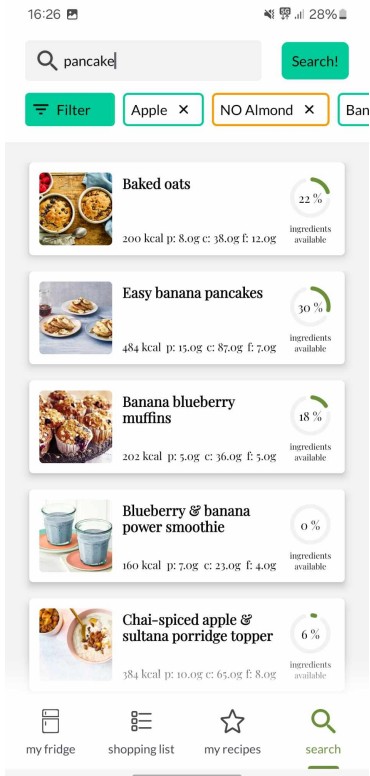

Rysunek 1: Widok wyszukiwania przepisów.

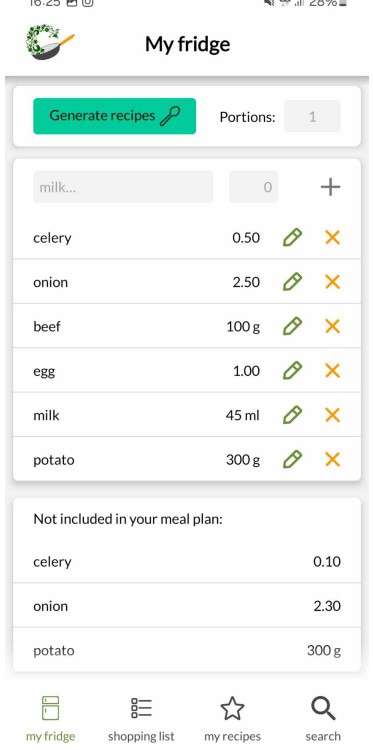

Rysunek 2: Widok posiadanych produktów.

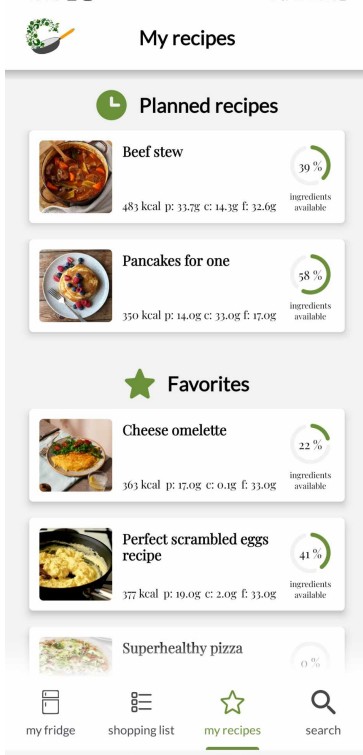

Rysunek 3: Widok zapisanych przepisów.

## 3.1    Zarządzanie posiadanymi produktami

Aby umożliwić obsługę pozostałych funkcjonalności aplikacji, potrzebny jest system efektywnego wprowadzania i przeglądania danych o posiadanych produktach spożywczych. Użytkownik ma możliwość wprowadzania informacji o nowym zakupionym produkcie, a także edycji ilości już posiadanego produktu lub całkowitego usunięcia go z listy. W tym procesie użytkownika wspierają sugestie tekstowe. Istnieje też opcja wybrania z listy wszystkich dostępnych produktów. Dodatkowo jednostka jest automatycznie dobierana do produktu i wyświetlana użytkownikowi, a wielokrotne wprowadzanie tego samego produktu powoduje zagregowanie dotychczas wprowadzonych wartości w jednym wierszu. Ponieważ trzymanie się wytycznych dotyczących zdatności do spożycia nie jest istotną potrzebą przedstawicieli grupy docelowej, ten typ informacji został pominięty w aplikacji. Pozwala to ograniczyć ilość wprowadzanych i wyświetlanych danych oraz pozostawia swobodę oceny zdatności produktów użytkownikowi. Aplikacja w prosty sposób prezentuje wszystkie posiadane produkty, dzięki czemu zmniejsza się możliwość wystąpienia sytuacji, w której użytkownik zapomni o posiadaniu produktu postawionego np. z tyłu lodówki bądź na dnie szafki. Użytkownik ma także dostęp do informacji o produktach, których nie zaplanował jeszcze wykorzystać.

## 3.2    Generowanie propozycji przepisów

Najbardziej pomocną funkcjonalnością aplikacji jest opcja generowania spersonalizowanych propozycji przepisów. Aplikacja ocenia przepisy pod kątem możliwości wykonania ich za pomocą wyłącznie posiadanych produktów i zwraca wynik procentowy, gdzie 100% oznacza, że użytkownik posiada wszystkie potrzebne produkty w ilości równej lub większej niż wymagana, a 0% oznacza, że użytkownik nie posiada żadnego potrzebnego produktu. Odpowiada to na kilka potrzeb docelowego użytkownika. Dobranie przepisów na podstawie posiadanych produktów zwiększa prawdopodobieństwo zużycia ich przed upływem okresu zdatności do spożycia, zwłaszcza w przypadku zapomnianych i mało widocznych w świecie rzeczywistym produktów. Wyświetlenie obiektywnej oceny zachęca do zużywania posiadanych produktów i zmniejsza wpływ chwilowych zachcianek na decyzję. Aplikacja pozwala na wygenerowanie propozycji na dowolną liczbę porcji i wyświetla przeskalowaną treść przepisu. Dzięki temu użytkownik może przygotować odpowiednią ilość jedzenia. Zwiększona widoczność nieznanych przez użytkownika przepisów i zmniejszenie liczby opcji do 10 propozycji zachęca do próbowania nowych dań i prowadzenia zróżnicowanej diety, jednocześnie skracając czas potrzebny na planowanie posiłków.

## 3.3    Funkcjonalności związane z przepisami

Użytkownik może dodać przepis do ulubionych. Powoduje to zapisanie na urządzeniu danych potrzebnych do wyświetlenia przepisu, co pozwala na łatwy powrót do niego nawet w przypadku braku połączenia z internetem. Użytkownik może także oznaczyć przepis jako zaplanowany do wykonania w najbliższym czasie. Spowoduje to zapisanie danych i zarezerwowanie potrzebnych składników. Podczas generowania przepisów zostaje obliczony przyszły stan produktów, pomniejszony o sumaryczną ilość wymaganą przez wszystkie zaplanowane przepisy. Umożliwia to planowanie kilku posiłków na raz bez potrzeby edytowania ilości posiadanych produktów. Po przygotowaniu danego przepisu użytkownik może wybrać opcję oznaczenia przepisu jako wykonanego. Wszystkie posiadane produkty wymagane przez przepis zostają wtedy pomniejszone o zużytą ilość lub usunięte, a przepis zostaje usunięty z listy zaplanowanych przepisów, jeśli był tak oznaczony.

## 3.4    Lista zakupów

Użytkownik ma łatwy dostęp do generowanej automatycznie listy zakupów. Pozycje na niej obliczane są na podstawie zaplanowanych przepisów i posiadanych produktów, a informacja o nich jest dostępna bez połączenia z internetem, aby można było z niej korzystać w sklepie. Użytkownik ma opcję zaznaczania i odznaczania pozycji na liście, a po wykonaniu zakupów może jednym kliknięciem zaktualizować listę posiadanych produktów o zakupioną ilość. W ten sposób użytkownik jest zachęcany do kupowania tylko potrzebnych produktów, a przy regularnym i rzetelnym używaniu aplikacji rzadko występuje potrzeba ręcznego wprowadzenia ilości zakupionych produktów.

## 3.5    Wyszukiwanie przepisów

W przypadku, w którym użytkownik posiada częściową wiedzę o tym, co chce ugotować, może skorzystać z opcji wyszukiwania przepisów. Podstawowe informacje, takie jak lista składników i sposób przygotowania, wzbogacone są o informacje o tym, ile produktów potrzebnych do przygotowania przepisu użytkownik już posiada i w jakiej ilości. Składniki odżywcze są w widoczny sposób oznaczone w zależności od tego,

czy mieszczą się w zalecanych normach, aby ułatwić użytkownikowi podejmowanie zdrowszych decyzji. Dostępne jest także filtrowanie przepisów w zależności od występowania w nich wybranych produktów, co ułatwia znalezienie odpowiedniego przepisu, gdy określone produkty niedługo przestaną być zdatne do spożycia. Użytkownik ma też opcję odfiltrowania przepisów, w których występują wskazane produkty, aby uniknąć alergenów lub mniej lubianych produktów, których kupno wiąże się z większym ryzykiem wyrzucenia ich w przyszłości.

## 3.6  Testy z udziałem użytkowników

### 3.6.1  Charakterystyka ankietowanych

Zaproponowane rozwiązanie powinno być proste w obsłudze i odpowiadać na problemy użytkowników, dlatego przeprowadziliśmy serię testów z udziałem użytkowników. W badaniu wzięło udział 12 osób należących do grupy docelowej, czyli młodych osób pracujących. Najpierw uczestnicy mieli możliwość swobodnej eksploracji aplikacji MealMaster bez żadnych instrukcji. Zostali wtedy poproszeni o jednoczesne opowiadanie o swoich wrażeniach. Następnie dostali zestaw zadań do wykonania w aplikacji, które przedstawiały przykładowe scenariusze użycia. Na koniec wszyscy uczestnicy wypełnili ankietę podsumowującą ich doświadczenia.

### 3.6.2  Wyniki testów

Aplikacja MealMaster okazała się intuicyjna. Uczestnicy poprawnie rozpoznawali znaczenie elementów interfejsu i znajdywali potrzebne im funkcjonalności. Poproszeni o zaplanowanie posiłku, chętnie korzystali z opcji generowania propozycji. W większości przypadków potrafili wybrać przepis, który odpowiadał ich potrzebom i byli zadowoleni z jakości tworzonych w ramach badań jadłospisów na kilka dni. Potrafili także ocenić, czy danie jest zdrowe, na podstawie przedstawionych im danych. Dotyczy to także uczestników, którzy nie byli zainteresowani zdrowym odżywianiem, jednak zaznaczali, że bez pytania od moderatora nie zwracali uwagi na składniki odżywcze.

Uczestnicy najlepiej odbierali listę zakupów i generowanie propozycji przepisów. Wskazywali na sytuacje, w których aplikacja pomogłaby im dzięki tym funkcjonalnościom i oceniali je jako bardzo pomocne. Proponowali także możliwe usprawnienia, które dodatkowo wzbogaciłyby ich doświadczenie. Chociaż uczestnikom podobało się, że lista zakupów jest auto-generowana, opisywali też, jak ważna jest dla nich możliwość jej edycji. Generowane propozycje były natomiast mało zróżnicowane, gdy użytkownik posiadał małą liczbę produktów, ale w dużej ilości. Zdarzało się także, że użytkownik posiadał produkt, który nie był składnikiem żadnego z przepisów.

Uczestnicy wskazali także na potrzebne usprawnienia innych funkcjonalności. Podczas wprowadzania produktów do aplikacji nie byli pewni, czy wpisane wartości były poprawne. Dezorientujący był też brak wyświetlania jednostki w przypadku wprowadzania produktów liczonych w sztukach. Uczestnicy chcieli wyszukiwać przepisy na podstawie ich produktów, jednak znalezienie odpowiedniego na liście dostępnych filtrów było dla nich zbyt pracochłonne. Jednocześnie chcieli mieć więcej opcji filtrowania, na przykład po typie diety (wegetariańska, bezglutenowa), typie posiłku (śniadaniowe, obiadowe, deserowe) lub rodzaju potrawy (makaron, zapiekanka). Uczestnikom podobała się możliwość generowania propozycji na konkretną liczbę porcji, wskazywali jednak na potrzebę skalowania przepisu w innych miejscach aplikacji, np. przy wyszukiwaniu przepisów lub na liście już zaplanowanych przepisów.

Było też kilka funkcjonalności, które uczestnicy nie zrozumieli w pełni. Oznaczenie kolorami składników odżywczych było pozytywnie oceniane, jednak nie wszyscy użytkownicy rozumieli znaczenie koloru pomarańczowego, użytego w aplikacji do oznaczenia składników odżywczych wychodzących poza zalecane normy. Analogicznie, wszyscy uczestnicy używali opcji planowania przepisów i opisywali jej użyteczność. Nie wiedzieli jednak, że wpływa ona na generowanie propozycji, przez co zazwyczaj wybierali kilka przepisów z jednej puli rekomendacji, zamiast generować nowe propozycje po zaplanowaniu przepisu.

### 3.6.3  Testy na prawdziwych danych

W trakcie testów uczestnicy zostali poproszeni o wprowadzenie do aplikacji posiadanych przez siebie produktów i zaplanowanie posiłków na 5 dni zgodnie ze swoimi preferencjami odnośnie przyrządzanych dań i częstotliwości gotowania. Uczestnicy byli zadowoleni z wyników. Oceniali jadłospisy jako bardziej zróżnicowane niż zazwyczaj i podkreślali łatwość w planowaniu. Niestety, uczestnicy deklarowali, że bez użycia aplikacji zakupiliby mniej produktów, ale kosztem jakości i różnorodności dań.

# 4 WNIOSKI

## 4.1 Wnioski ogólne

Aplikacja MealMaster została wysoko oceniona pod względem doświadczeń użytkownika. Według ankiet rozesłanych po testach z użytkownikami średnia ocena łatwości w zrozumieniu funkcjonalności wyniosła 4.75/5, a ocena łatwości w obsłudze - 4.42/5. W trakcie testów zostały ujawnione obszary, które można jeszcze poprawić. Nasz zespół postanowił usprawnić wprowadzanie produktów do aplikacji tak, aby użytkownik lepiej rozumiał, czy wprowadzone przez niego dane są poprawne. Pola tekstowe z poprawnym stanem zostaną otoczone zieloną ramką, a jednostka będzie widoczna także w przypadku produktów liczonych w sztukach. Z powodu zbliżającego się terminu zakończenia projektu, nie zostały zaplanowane dalsze usprawnienia na podstawie testów.

W kwestii celów biznesowych, ankietowani ocenili aplikację wysoko zarówno pod względem pomocy w podejmowaniu świadomych decyzji żywieniowych (4.25/5) jak i w minimalizowaniu ilości marnowanej żywności (4.75/5). Drugi wynik nie pokrywa się jednak z wynikami testów na prawdziwych danych, z których wynikało, że w przypadku używania aplikacji użytkownicy potrzebowali zakupić więcej produktów niż zazwyczaj. Test nie odzwierciedlał dobrze rzeczywistych warunków, w których użytkownik używałby aplikację przez wiele dni. W takim przypadku mogłoby okazać się, że po wstępnych zakupach zwiększających różnorodność posiadanych produktów, aplikacja proponowałaby przepisy możliwe do wykonania na bazie prawie wyłącznie posiadanych już produktów. Do rzetelnej oceny aplikacji pod tym kątem potrzebne są długoterminowe testy z grupą kontrolną.

## 4.2 Potencjalne kierunki rozwoju

Zrealizowany projekt cechuje się znacznym potencjałem dalszego rozwoju, co wynika zarówno z analizy wyników przeprowadzonych badań jakościowych, jak i identyfikacji obszarów wymagających usprawnienia. Wprowadzenie nowych funkcjonalności oraz usprawnień może znacząco podnieść wartość użytkową aplikacji oraz zwiększyć jej atrakcyjność wśród odbiorców. Poniżej przedstawiono propozycje kluczowych kierunków rozwoju, uwzględniające zarówno potrzeby użytkowników, jak i kwestie związane z długoterminowym finansowaniem projektu:

### 4.2.1 Personalizacja pod kątem diety

Dodanie opcji filtrowania przepisów zgodnie z preferencjami dietetycznymi użytkownika, np. eliminacja dań zawierających gluten, laktozę lub mięso. Funkcjonalność ta mogłaby zostać także zintegrowana z algorytmem generowania przepisów, co pozwoliłoby na jeszcze większą personalizację.

### 4.2.2 Dodawanie przepisów przez użytkowników

Umożliwienie użytkownikom rozszerzania bazy przepisów poprzez dodawanie własnych propozycji. Taki przepis mógłby być widoczny tylko dla użytkownika albo udostępniony całej społeczności. Takie rozwiązanie obniżyłoby koszty związane z rozbudową bazy danych i jednocześnie zapewniłoby jej większą zgodność z realnymi oczekiwaniami odbiorców. Dodatkowo aby zrealizować ten cel, konieczne byłoby wprowadzenie automatycznej moderacji treści w celu zapobiegania publikacji nieodpowiednich materiałów, przy jednoczesnej minimalizacji kosztów operacyjnych.

### 4.2.3 Śledzenie dat przydatności składników

Wprowadzenie funkcji automatycznie zapisującej datę dodania posiadanych produktów do bazy danych oraz uwzględniającej średni czas ich przydatności do spożycia. Ten mechanizm mógłby zostać wykorzystany do lepszego pozycjonowania przepisów zawierających składniki, które należy zużyć w pierwszej kolejności.

### 4.2.4 Uatrakcyjnienie interfejsu

Wzbogacenie aplikacji o interaktywne animacje oraz inne elementy wizualne mające na celu poprawę doświadczeń użytkowników. Przyjazny interfejs i pozytywne emocje związane z korzystaniem z aplikacji mogą przyczynić się do zwiększenia lojalności odbiorców.

### 4.2.5 Monetyzacja projektu

Wprowadzenie modelu zarabiania na aplikacji poprzez integrację reklam wyświetlanych pomiędzy przepisami. Dla użytkowników preferujących korzystanie z aplikacji bez reklam możliwe byłoby wprowadzenie subskrypcji premium.

### 4.2.6 Wykorzystanie technologii AI

Personalizacja przepisów przy użyciu zaawansowanych algorytmów sztucznej inteligencji. Funkcjonalność ta, dostępna dla użytkowników premium, umożliwiałaby generowanie przepisów dostosowanych do indywidualnych preferencji na podstawie analizy historii wcześniej wybranych przepisów. Dodatkowo uwzględniano by kontekstowe czynniki, takie jak dzień tygodnia, pora dnia, pora roku czy preferowana dieta.

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
