# OpenReview forum: "MealMaster - aplikacja mobilna wspierająca zbilansowaną dietę oraz walkę z marnowaniem żywności"
_pwr.edu.pl/Wrocław_University_of_Science_and_Technology/2024/ZPI_Day — Wrocław University of Science and Technology 2024 ZPI Day Submission_

### Official Review · Reviewer_nHAD · 2024-12-03
**Well described application, testing is average at best**

**Confidence:** 5
**Significance Of Results:** 4
**Overall Quality:** 4

**Compliance With Template:**

4: High Quality – The article contains all the required sections, which are well-written and substantively correct, although minor errors or shortcomings may be present. The overall structure is clear and coherent.

**Description Of Results:**

4: High Quality – The results are described in detail and supported by usage examples or evaluations. The description is reliable but may lack full depth of analysis.

**Feedback On Consistency:**

The description is internally consistend, the presentation is correct (even if long, compared to intended 4 pages), the flow is logical and consistent.

**Potential For Development:**

There are several possible directions of future work outlined.

**Project Nature Evaluation:**

The project is a correct engineering work of practical utility and may well be deployed as (non) commercial application. One possible problem is scalability of the solution, which was not tested (with correct approach, for this type of app it shouldn't be a problem). User tests were conducted, but are not well described here.

**Technical Language Precision:**

4: High Quality – The language is appropriate for a technical report. Terminology is used correctly, and statements are precise, with only minor shortcomings that do not affect the overall clarity.

---

### Official Review · Reviewer_iC6U · 2024-12-04
**MealMaster**

**Confidence:** 5
**Significance Of Results:** 4
**Overall Quality:** 5

**Compliance With Template:**

5: Very High Quality – The article contains all the required sections, which are written in a very detailed, clear, and error-free manner. The structure is professional and meets expectations, and the content adheres to the highest substantive and formal standards.

**Description Of Results:**

5: Very High Quality – The results are described in detail, clearly and comprehensively, supported by thorough evaluation, analysis, and convincing usage examples. The description meets the highest substantive standards.

**Feedback On Consistency:**

Kompletny artykuł, zawierający również wyniki testów na użytkownikach końcowych. Napisany na odpowiednim poziomie szczegółowości z prawidłowymi powołaniami literatury. Zadawalająca innowacyjność propozycji oraz dowody jej realizacji.

**Potential For Development:**

Przedstawiono kierunki rozwoju. Aplikacja ma potencjał wdrożenia.

**Project Nature Evaluation:**

Opisano decyzje w zakresie technologii z uzasadnieniem.

**Technical Language Precision:**

5: Very High Quality – The language is entirely appropriate for a technical report. All terms are used correctly and precisely, and the style is professional, clear, and coherent, without any errors or ambiguities.

---

### Official Review · Reviewer_ppyt · 2024-12-06
**MealMaster - project abstract review**

**Confidence:** 4
**Significance Of Results:** 5
**Overall Quality:** 5

**Compliance With Template:**

5: Very High Quality – The article contains all the required sections, which are written in a very detailed, clear, and error-free manner. The structure is professional and meets expectations, and the content adheres to the highest substantive and formal standards.

**Description Of Results:**

4: High Quality – The results are described in detail and supported by usage examples or evaluations. The description is reliable but may lack full depth of analysis.

**Feedback On Consistency:**

The motivation and significance of the project, the analysis, presentation of results, and conclusions are fully consistent and logical. The project of the application itself could be slightly more detailed described but it doesn’t noticeably affect the overall assessment.

**Potential For Development:**

Possibilities for further work and practical applications are very well described in the work. The potential of application is high. The first step is to adopt product expiration date of products, what was omitted in application.

**Project Nature Evaluation:**

The project very well fulfills requirements for the utility and practical application of the implemented application. It seems that characteristics of an engineering work also deserve very good assessment, however some key techniques of application itself are not described (I mean key technical solutions), which nevertheless must had been used and implemented. Taking into account product expiration dates should be one of the first implemented features in the application.

**Technical Language Precision:**

5: Very High Quality – The language is entirely appropriate for a technical report. All terms are used correctly and precisely, and the style is professional, clear, and coherent, without any errors or ambiguities.

---

### Decision · Program_Chairs · 2024-12-10

Accept (Oral)